# Circular Learning Provides Biological Plausibility

**Mohammadamin Tavakoli**
Computing and Mathematical Sciences
California Institute of Technology
`amint@caltech.edu`

**Ian Domingo**
Computer Science Department
University of California, Irvine
`idomingo@uci.edu`

**Pierre Baldi**
Computer Science Department
University of California, Irvine
`pfbaldi@uci.edu`

## Abstract

Training deep neural networks in biological systems is faced with major challenges such as scarce labeled data and obstacles for propagating error signals in the absence of symmetric connections. We introduce Tourbillon, a new architecture that uses circular autoencoders trained with various recirculation algorithms in a self-supervised mode, with an optional top layer for classification or regression. Tourbillon is designed to address biological learning constraints rather than enhance existing engineering applications. Preliminary experiments on small benchmark datasets show that Tourbillon performs comparably to models trained with backpropagation and may outperform other biologically plausible approaches. The code and models are available at `https://github.com/IanRDomingo/Circular-Learning`.

## 1   Introduction

Decades of machine learning have taught us that gradient descent is the sole effective optimization method in high-dimensional spaces. Other strategies, like random search, are bound to fail. Backpropagation, the algorithm behind gradient computation in artificial neural networks, has been incredibly successful. It powers advancements in Artificial Intelligence, from protein folding (e.g., AlphaFold (1)) to natural language understanding and generation (e.g., GPT-4 (2; 3)). Backpropagation efficiently computes the gradient in a network with $W$ weights using $O(W)$ operations. Considering that at least $O(W)$ operations are necessary to adjust $W$ synapses, backpropagation demonstrates optimal efficiency. Consequently, if learning is viewed as an optimization problem in a high-dimensional space of synaptic weights, this suggests that the brain likely employs learning algorithms based on gradient computation, either exact or approximate. Yet there are several well-known reasons in the literature why backpropagation is implausible in biological systems (4; 5; 6; 7; 8). Thus, in short, we hypothesize that biological systems must strive to approximate gradient descent methods without being able to compute exact gradients by backpropagation. Here, we set out to propose a plausible strategy for achieving this goal.

To begin, we outline the key factors that make existing neural architectures biologically implausible: **1) Symmetry of Connections** (weight transport): Backpropagation requires precisely symmetric connections between the forward and backward passes. This constraint cannot be satisfied in a biological neural system and might be hard to realize in some physical neural systems. **2)Forward Nonlinearities** (F prime): Backpropagation relies on an exact memory in the backward pass of the nonlinearities applied in the forward pass, such as activation functions, to compute weight derivatives. However, there is no evidence supporting the existence of such precise memory in biological or physical neural systems. **3) Locality**: In a biological neural system, the learning rule for

adjusting synaptic weights must be local, i.e. it must rely solely on variables available locally, both in space (spatial locality) and time (temporal locality), at each synapse. **4) Clocked Computation**: In backpropagation, the forward and backward passes are manually clocked to compute activations and update weights. In contrast, in a biological system, neurons communicate stochastically, lacking the precise clocking mechanism observed in backpropagation. **5) Labeling**: Training classifiers rely on large amounts of labeled data for supervised learning. However, biological systems do not seem to have access to large amounts of labeled data. **6) Spike**: While biological neurons communicate using noisy spikes, artificial neurons typically communicate using deterministic analog values (with known exceptions such as dropout). **7) Distances**: The architecture of deep neural networks necessitates propagating signals over considerable neural distances in deep models, which can result in signal dilution and lead to distorted or unstable gradients. **8) Developmental Modularity**: Backpropagation in general, requires having a complete architecture in place before training can begin, which may not be realistic for biological systems undergoing development and other changes.

Several solutions have been suggested to try to address these problems, in isolation or small combinations, but no approach addresses all of them at once. Here we propose a neural architecture called Tourbillon and its training algorithms to address all the implausibility discussed above by combining different ideas, including stacked autoencoders, recirculation, and asynchronous training. We emphasize that the primary goal here is to address the obstacles listed above for biological (or neuromorphic) neural systems and not to derive a new architecture or algorithm that is practically useful for digital applications of deep learning.

## 2 Biological Plausibility

Several approaches have been proposed to address the biological implausibilities enumerated above. The most notable ones include Feedback Alignment (FA) (9; 10; 11; 12), Difference Target Propagation (DTP) (5), Stacked Autoencoders (13; 14), and the Forward-Forward (FF) algorithm (15). However, each of these methods addresses only a limited subset of the biological implausibilities (Section A.1 and Table 1). Self-supervised learning, in particular stacked autoencoders, provides one way of addressing the data labeling issue. However, standard autoencoders suffer from several other issues which we now address.

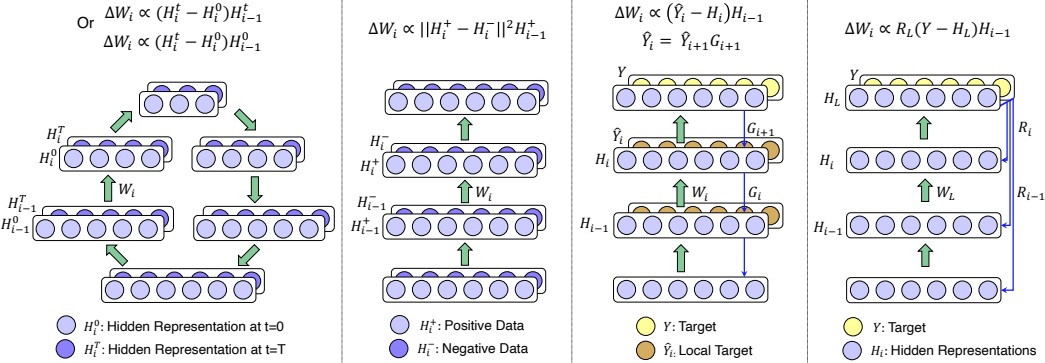

Figure 1: From left to right: Recirculation, forward forward, difference target propagation, (direct) feedback alignment. The learning rule for each model is written at the top of the architecture schematics.

Table 1: A comparison of physical plausibility between different neural architectures from a biological standpoint. ✘, ⊛, and ✔ correspond to no plausibility, partial plausibility, and full plausibility, respectively.

| | W Transport | F prime | Locality | Clocked | Labeling | Spike | Distance | Modular |
|---|---|---|---|---|---|---|---|---|
| Backpropagation | ✘ | ✘ | ✘ | ✘ | ✘ | ✘ | ✘ | ✘ |
| Feedback Alignment (FA) | ✔ | ✔ | ✘ | ✘ | ✘ | ✘ | ✘ | ✘ |
| Direct Feedback Alignment (DFA) | ✔ | ✔ | ⊛ | ✘ | ✘ | ✘ | ✔ | ✘ |
| Difference Target Propagation (DTP) | ⊛ | ✘ | ✘ | ✘ | ⊛ | ✘ | ✔ | ✘ |
| Stacked Autoencoders | ✔ | ✔ | ⊛ | ✘ | ✔ | ✘ | ✔ | ⊛ |
| Forward Forward (FF) | ✔ | ✔ | ✔ | ✘ | ⊛ | ✘ | ✔ | ✘ |
| **Tourbillon** | ✔ | ✔ | ✔ | ✔ | ⊛ | ✘ | ✔ | ✔ |

**Circular Autoencoders.** In a standard feed-forward autoencoder (AE), the data itself provides the targets (self-supervised learning). The data and hence the targets are available in the input layer. However, they are not available in the output layer, in the sense that they are not physically local (spatial-locality) to the output layer. This problem is addressed in circular autoencoders (CAE) (16) where the output layer is physically equal (or physically adjacent) to the input layer (Figure 1). With the circular layout, targets and errors can be computed at the level of the input/output layer.

**Recirculation Algorithms.** Standard backpropagation, or even FA, of these targets, would require a channel (wires) running backward from the output layer to the hidden layer. However, because of the circular layout, it is possible to use the forward connections to propagate target and error information during learning. This is the fundamental idea behind recirculation, a family of algorithms for training CAEs that do not require backward connections (17; 18; 7).

Consider a CAE with layers numbered from $0$ to $L$, where $0$ corresponds to the input layer. We use the index $t$ to denote different cyclic passes through the autoencoder, with the first pass indexed by $t = 0$. After the first pass, one can *locally* compute the error $T - H_L^0$, where $T$ is the target located at the input layer. This error could be used to train the top layer of the CAE by gradient descent, and then train the other layers by using a form of random backpropagation where the error signal is obtained by propagating the error $T - H_L^0$ using the forward weights of the CAE. This however requires propagating two different kinds of signals, activities, and errors, through the CAE. Thus rather than recirculating the error, a more uniform approach can be obtained by recirculating activities. If $H_i^t$ denotes the activation of layer $i$ during the forward pass indexed by $t$, the main idea behind the recirculation family of algorithms is to use $H_i^t$ as the target for the output $H_i^{t'}$ taken at a later time $t'$ to produce the post-synaptic term for the weight update. The intuition is that the data may become increasingly corrupted as it is being recycled, thus earlier pass serve as targets for later passes. Different variations can be obtained, by varying, for instance, the post- and pre-synaptic terms. Equation 1 describes the update rule of the weights in a circular autoencoder.

$$\Delta W_i = \eta (H_i^t - H_i^{t'})^{post} (H_{i-1}^t)^{pre} \tag{1}$$

This rule follows a Hebbian-product form, resembling backpropagation but with a postsynaptic recirculation error, denoted as $[H_i^0 - H_i^1]^{post}$. This error term is both spatially and temporally local, assuming that consecutive passes through the circular autoencoder fall within the proper time window. In the input layer, the vector $H_0^0$ represents the input data, including the targets for an autoencoder. Consequently, the recirculation learning equation for the top layer of weights is identical to backpropagation. Although in this work we are not using spiking neurons, such learning rules are closely related to the concept of spike time-dependent synaptic plasticity (STDP) (19). STDP Hebbian or anti-Hebbian learning rules have been proposed using the temporal derivative of the activity of the post-synaptic neuron (20) to encode error derivatives.

## 3 Tourbillon: A CAE Stack

We propose the Tourbillon architecture as a stack of circular autoencoders, capped by a classification or regression layer connecting the hidden representation of the top circular autoencoder and the output layer. Each circular autoencoder has an encoder and decoder components. The hidden layer that is shared by the encoding and decoding components is called the hinge layer. In the stack, the hinge layer of the $i$th circular autoencoder becomes the input layer of the $i+1$th circular autoencoder (Figure 2 (b)). The Tourbillon architecture addresses the issues of target labels and spatial locality. With the recirculation algorithms, it also addresses the issues of weight transport, forward non-linearities, temporal locality, and distances. Using a novel training algorithm, we set out to address issues of clocking and modularity.

**Asynchronous Training.** To fully address modularity and provide non-clocked computations, we consider asynchronous training. In this case, each CAE can be viewed as a "spinning wheel" and these wheels can spin independently of each other. At any random time, a CAE may elect to recirculate whatever happens to be in its input layer and adapt its synapses accordingly. The algorithm for asynchronous training is given in the Appendix.

## 4 Experiments and Results

We begin by training CAEs and investigate the effects of various parameters, including the number of cycles ($t$), the CAE size (i.e., the number of hidden layers in the CAE except the input and output

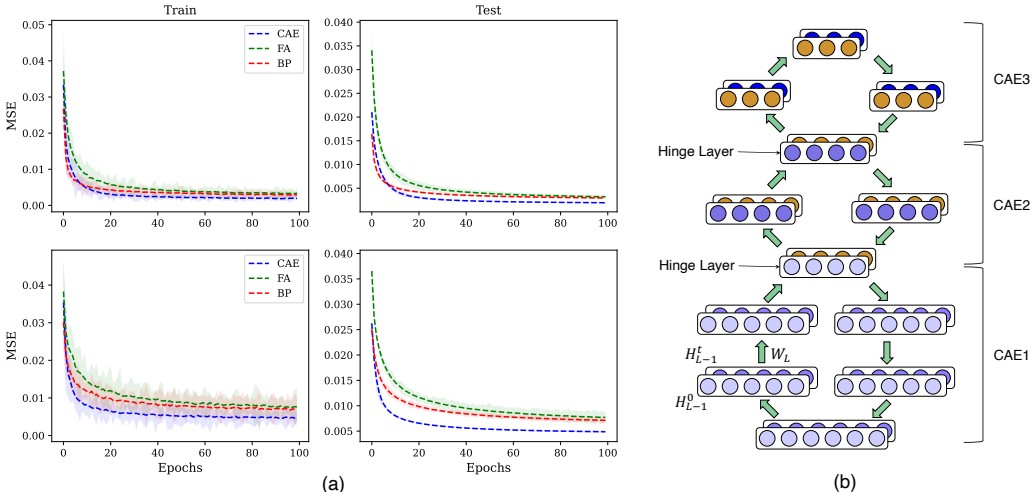

Figure 2: (a): Train and test loss of three autoencoders trained with backpropagation (BP), feedback alignment (FA), and recirculation (CAE) on MNIST (top row) and Fashion MNIST (bottom row). Each line shows the mean of five runs, with shaded areas indicating standard deviation. (b) Tourbillon architecture with a stack of three circular autoencoders (CAE) trained by recirculation.

Table 2: The mean-squared reconstruction loss of CAEs with different cycles and CAE sizes trained on MNIST and Fashion MNIST. Each number is the mean of five distinct runs. The top results are in boldface.

| CAE size | MNIST Cycles | | | Fashion MNIST Cycles | | |
|---|---|---|---|---|---|---|
| | 1 | 2 | 3 | 1 | 2 | 3 |
| 1 | 0.0099 | 0.0093 | **0.0090** | 0.0132 | 0.0124 | **0.0123** |
| 3 | 0.0165 | 0.0154 | 0.0151 | 0.0204 | 0.0198 | 0.0204 |

layers), and different learning rules (Equation 1), on training dynamics. Then, using the best set of these parameters, we develop and train several Tourbillon architectures, both with and without a top classifier layer, using different stack depths and training algorithms. The goal of these experiments is not to outperform existing deep learning models but to show that Tourbillon architectures can learn complex tasks while satisfying the plausibility constraints. Similar to recently proposed plausible architectures (15), we use relatively small datasets and models, leaving the scaling up to future studies. Details on the hyperparameters, hardware, and CAE implementation are in the appendix and the gitHub repository.

## 4.1 Training Tourbillon CAEs

We train CAEs using the learning rule in Equation 1. We optimize each architecture using a mean-squared reconstruction loss. In all experiments, we use symmetric CAEs, where the number of hidden layers in the encoder and decoder are equal. To satisfy distance plausibility, we use CAEs with a small number of hidden layers (CAE size = one and three). Additionally, to maintain the temporal locality of the variables, we limit the number of cycles ($t$) to less than four. We train CAEs with fully connected layers for the MNIST and Fashion MNIST datasets. Table 2 displays the reconstruction loss on the test datasets. Notably, using a CAE size of one and one cycle ($t = 1$) yields the lowest testing loss. This corresponds to the greatest level of spatial and temporal locality. Using the best values for the CAE size and number of cycles, we further show the viability of training CAEs with the learning rule above. We compare the mean-squared loss of the trained CAE with the same autoencoder trained with backpropagation, FA, and DFA. Figure 2 (a) shows the training and test error curves for the MNIST and Fashion MNIST datasets. Our results show that recirculation achieves comparable, and possibly superior, reconstruction errors compared to backpropagation and FA.

Table 3: The mean-squared reconstruction loss of CAEs with different depths on MNIST, Fashion MNIST.

| | MNIST Stack Depth | | | Fashion MNIST Stack Depth | | |
|---|---|---|---|---|---|---|
| | 2 | 3 | 4 | 2 | 3 | 4 |
| | **0.0088** | 0.0190 | 0.0251 | **0.0141** | 0.0290 | 0.0426 |

## 4.2 Tourbillons With Various Depth

We construct stacks of two, three, and four compressive CAEs and train them using asynchronous algorithms. Table 3 presents the reconstruction error of the stacks, indicating the depth and training algorithm applied to each CAE.

Experimenting with different learning rate schedules for each CAE in the stack reveals that decreasing the learning rates from the bottom to the top layers is crucial. The training algorithm's inherent randomness also leads to a higher test reconstruction error. However, the focus of this study is not on performance but on introducing a biologically plausible training algorithm and demonstrating its feasibility. After training the stacks, we add a top classifier layer. Based on Table 3, we use three CAEs for MNIST and Fashion MNIST. We conduct classification experiments to compare Tourbillon's performance with neural networks of similar architecture trained using backpropagation, FA, and DFA. Tourbillon outperforms FA and matches backpropagation, particularly in fully connected architectures. A key advantage of Tourbillon is its ability to leverage unlabeled data for unsupervised training of the stack, allowing the top classifier to be trained with less labeled data. This reduces reliance on labeled data, enhancing biological plausibility.

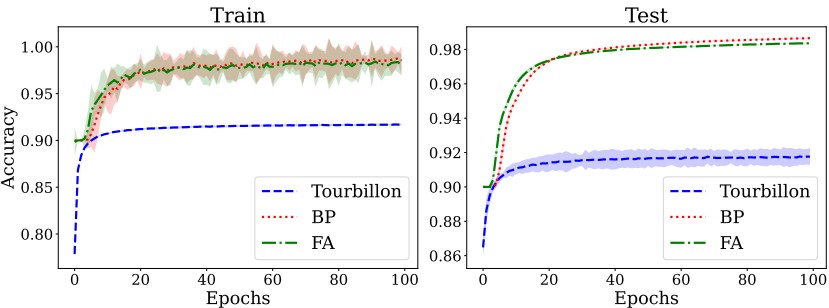

Figure 3: Train and test accuracy of three classifiers trained on MNIST dataset using backpropagation (BP), feedback alignment (FA), and Tourbillon. Each line corresponds to the mean of five distinct runs with the standard deviation shown as the shaded area.

## 5 Conclusion

Tourbillon represents a systematic approach toward addressing the major biological implausibilities in both structures and training algorithms of existing neural networks. In essence, it is a stack of circular autoencoders, each trained by recirculation at random times. Hence we chose the name Tourbillon (associated with turbulence in French) the turbulent topology of the architecture. Moreover, in horology, a tourbillon is an addition to the mechanics of a watch escapement to increase its accuracy. While we do not claim to have increased accuracy, we have shown that the Tourbillon approach shows similar performance to existing neural networks, at least on small datasets, while being more biologically plausible. In conclusion, Tourbillon serves as a framework to investigate the biological implausibilities of artificial neural architectures and aims to advance the field of biologically plausible deep learning. By addressing key implausibilities, Tourbillon opens up new possibilities for studying neural networks in accordance with biological principles.

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

# A   Appendix

In this appendix, we provide additional details regarding the algorithms and the experiments. All the experiments are conducted using a single NVidia Titan X GPU.

## A.1   Biological Plausibility

We describe the learning equation (weight update) using post- and pre-synaptic terms. For a forward weight $W_i$ at layer $i$, the backpropagation learning equation can be written as:

$$\Delta W_i = \eta B_i^{\text{post}} H_{i-1}^{\text{pre}}, \quad B_L^{\text{post}} = T - H_L, \tag{2}$$
$$B_{i-1}^{\text{post}} = F_{i-1}' \circ W_i^T B_i^{\text{post}}$$

where $\eta$ denotes the learning rate, $B_i^{post}$ denotes the postsynaptic backpropagated error at layer $i$, and $H_{i-1}^{pre}$ denotes the pre-synaptic activity. $F_{i-1}' \circ$ denotes the component wise multiplication by the vector of activation function derivatives in layer $i-1$. $T$ and $H_L$ are the targets and the activation at the top layer $L$. Note that in order to avoid further cluttering the notation, we omit the transpose sign for all the presynatpic terms throughout this document.

### A.1.1   Feedback Alignment

Feedback Alignment (FA) or Random Backpropagation (RBP) refers to a family of algorithms (9; 10; 11; 12) that address the weights transport problem by using non-symmetric, and usually random, weights in the backward pass as follows:

$$\Delta W_i = \eta B_i^{post} H_{i-1}^{pre}, \quad B_L^{post} = T - H_L, \tag{3}$$
$$B_{i-1}^{post} = F_{i-1}' \circ R_i B_i^{post}$$

where $R_i$ denotes the random fixed matrices (random backward channels) to fix the issue of weights transport.

While FA and its variants address the weight transport problem, by themselves they do not address the other problems. Among several flavors of FA (10; 21), Direct Feedback Alignment (DFA) (11), backpropagates the error signal obtained at the top layer to each of the lower layers independently using direct fixed random matrices. By this means, DFA can address the issues of spatial locality and distance implausibilities. The learning equation of DFA can be written as follows:

$$\Delta W_i = \eta B_i^{post} H_{i-1}^{pre}, \quad B_i^{post} = R_i(T - H_L) \tag{4}$$

A version with component-wise multiplication by the derivatives of the corresponding activation functions is also possible. Experiments reported in the literature suggest that FA and its variants do not work well with convolutional layers (22; 23; 24). A few methods have been proposed to address this apparent weakness of FA algorithms, however, most of them introduce more constraints and dependence on the forward pass which may make them less biologically plausible (23; 25; 26).

### A.1.2   Difference Target Propagation

Difference Target Propagation (DTP) (5) as another biologically plausible model, trains the weights using a local target at each layer $\hat{Y}_i$ that is propagated from the original target $Y$ to each of the lower layers using learnable weights $G_i$.

$$\Delta W_i = \eta \hat{Y}_i^{post} H_{i-1}^{pre}, \quad \hat{Y}_i = \hat{Y}_{i+1} G_{i+1} \tag{5}$$

The $G_i$s are trained in the forward pass to approximate the inverse of the forward operation at each layer. Propagating the target using $G_i$s at the top two layers is dependent on the backpropagation and weight transport (24). Also, the forward and backward passes through the network are completely clocked to learn $G_i$s. However, since the information flows through layers independently, the variables are local in space, thus, this architecture can address space-locality and distance implausibilities.

### A.1.3 Stacked Autoencoders

A well known approach to address the labeling issue is using a stack of autoencoders (13; 14), where each autoencoder learns to reproduce the hidden representation of the previous one in a self-supervised manner, allowing the stack to learn increasingly abstract representations without labels. Labels are only used to train the top layer in a supervised way, with the option to fine-tune all layers via backpropagation (27).

This approach also addresses the distance and developmental modularity issues since backpropagation within each autoencoder limits error gradients to short distances and allows training to begin before the entire architecture is complete. However, stacked autoencoders do not solve the locality and weight transport issues. Each autoencoder, being deep, requires backpropagation across at least two adaptive layers, necessitating a learning channel for error signals and symmetric weights to implement backpropagation.

### A.1.4 Forward Forward

The recently introduced Forward Forward algorithm (FF) (15), attempts to address the implausibility through a contrastive learning framework. Positive and negative data are fed through the network. Then the weights can be updated using a local target defined at each layer as follows:

$$\Delta W_i = \eta(||H_i^+ - H_i^-||^2)^{post} H_{i-1}^{pre} \tag{6}$$

FF uses variables that are local in space and can be assumed to be local in time (due to the short neural distance). Given the contrastive learning framework, it can be trained in a self-supervised manner, however, the computation remains heavily clocked for feeding positive and negative data one at a time.

### A.2 Training Tourbillon CAEs

Table 4 summarizes the parameters used for training the CAEs in Section 4.1 of the main article.

Table 4: CAE size refers to the number of hidden layers except for the input and output layers. The CAEs are symmetric, with an equal number of hidden layers in both the encoder and decoder.

| CAE size | MNIST and Fashion MNIST Hidden Layers Dim |
|:---:|:---:|
| 1 | 784-256-784 |
| 3 | 784-256-64-256-784 |

Additionally, all models were trained for 100 epochs with a batch size of 64. To optimize the activation function and learning rates, a grid search was conducted, resulting in the use of *tanh* activation function and a learning rate of 0.01 for the initial layers. Subsequently, a smaller learning rate of 0.001 was employed for the remaining fully connected layers across all architectures.

### A.3 Stacking CAEs With Various Depth and Training Algorithms

Here we explain the details of the experiments conducted in Section 4.2. Specifically, Table 5 summarizes the parameters used for stacking and training the CAEs.

Table 5: Depth refers to the number of CAEs used to construct the stack.

| Depth | Input and Hinge Layers Dim |
|:---:|:---:|
| 2 | (784,256)-(256,128) |
| 3 | (784,256)-(256,128)-(128,64) |
| 4 | (784,256)-(256,128)-(128,64)-(64,64) |

For the asynchronous training algorithm, we use a batch size of 64 and we train the entire stack for 3000 iterations. According to Algorithm 1, an iteration refers to feeding one batch of data through

**Algorithm 1:** Asynchronous training

**Input:** $T$: A stack of $m$ sequential circular autoencoders $T = CAE_m \circ ... \circ CAE_1$, $CAE_i = D_i \circ E_i(datasample)$, $data$: training data, $S$: steps
**for** $i = 1$ **to** $S$ **do**
    $1 \leq j = random \leq m$
    $h = E_{j-1} \circ ... \circ E_1$
    $circulation(CAE_j, h)$

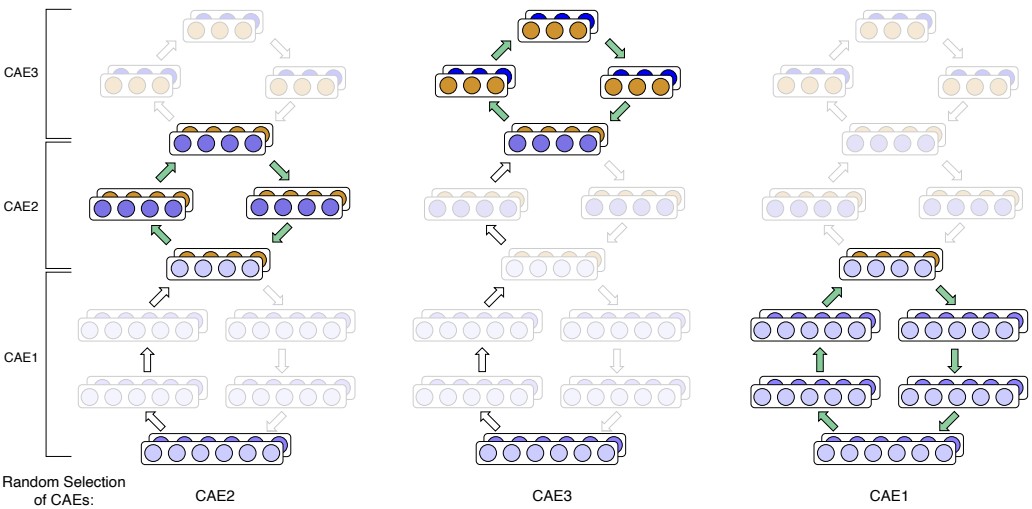

Figure 4: Random phases of the asynchronous training. Each time, one CAE is selected randomly and trained by recirculating the information.

the stack and updating the weights of one CAE within the stack. We provide a pseudocode of the asynchronous training in Algorithm 1. To further enhance clarity, we have depicted the schematics of the asynchronous training algorithm in Figure 4.

To evaluate the performance of the Tourbillon architecture when adding the top classification layer, we conducted a comparison with similar architectures trained using backpropagation, FA, and DFA. Figure 3 presents the results of this experiment specifically for the Tourbillons, trained sequentially with a stack of three fully connected CAEs using the Fashion MNIST dataset.

