# OpenReview forum: "Circular Learning Provides Biological Plausibility"
_NeurIPS.cc/2024/Workshop/UniReps — UniReps_

### Official Review · Reviewer_LvDJ · 2024-10-01
**Review of Circular Learning Provides Biological Plausibility**

**Rating:** 7
**Confidence:** 4

**Review:**

- The first paragraph seems unnecessarily argumentative. For instance, "Decades of machine learning have taught us that gradient descent is the **sole** effective optimization method in high-dimensional spaces" (emphasis mine). I agree that gradient descent is highly effective, but maybe it isn't the only optimization method that is so effective. Maybe we use it because it works and thus we haven't needed to find a better alternative. I don't think anything would be lost by instead saying "Decades of machine learning have taught us that gradient descent is an extremely effective optimization method in high-dimensional spaces"
- nit: Why are there no citations for " Feedback Alignment (FA), Difference Target Propagation (DTP), Stacked Autoencoders, and the Forward-Forward (FF) algorithm."
- nit: Fix the incorrect figure reference on Line 101 "(Figure ??)."
- Overall, I think this is an interesting idea. Its possible advantages aren't well demonstrated empirically, but I think it's worth discussing at the workshop.

---

### Official Review · Reviewer_jJva · 2024-10-05
**The authors introduce Tourbillon, a biologically plausible architecture using circular autoencoders trained asynchronously with different recirculation algorithms in a self-supervised manner. They provide preliminary results on the MNIST and Fashion MNIST datasets.**

**Rating:** 6
**Confidence:** 3

**Review:**

The authors introduce Tourbillon, a biologically plausible architecture using circular autoencoders trained asynchronously with different recirculation algorithms in a self-supervised manner. They provide preliminary results on the MNIST and Fashion MNIST datasets.

Strengths:
- The paper is well-written, with clear motivation and is easy to follow.
- Tourbillon is more biologically plausible than other bio-compatible methods.

Weaknesses:
- Several inconsistencies and errors are present, such as:
   1) The caption of Table 2 mentions CIFAR10 results, which are not included in the table.
   2) Line 101 references a figure, but it’s unclear which one.

- The experiments use small networks on simple datasets. While the authors acknowledge that scaling up is not their current goal, it would still be interesting to explore how this approach could perform in larger architectures. However, I don't consider this a major drawback.

---

### Official Review · Reviewer_tB9g · 2024-10-06
**The paper should provide more detailed explanations of Tourbillon's approach to biological implausibilities, a comparison with a standard baseline model, and correcting a missing figure number in a specific section.**

**Rating:** 5
**Confidence:** 3

**Review:**

Improvements:
1. Please provide more detailed explanations of how Tourbillon addresses each biological implausibility.
2. The paper does not directly compare the Tourbillon with a standard baseline model. Please include the baseline model.
3. In Section "3 Tourbillon: A CAE Stack", the figure number is missing in line 101. Please attach the correct figure number.

---

### Decision · Program_Chairs · 2024-10-10

**Decision:**

Accept

**Comment:**

In light of the positive reviewers' feedback and relevancy of the submission, we are pleased to accept this paper for presentation at UniReps 2024. We kindly ask the authors to incorporate the reviewers' suggestions and feedback in the final camera-ready version of the manuscript.